# Surface reflectance biases in XCH$_4$ retrievals from the 2.3 µm band are enhanced in the presence of aerosols

Peter Somkuti[1,2], Gregory McGarragh[3], Christopher O'Dell[3], Antonio Di Noia[4,5,6], Leif Vogel[4,5,7], Sean Crowell[8], Lesley Ott[2], and Hartmut Bösch[4,5,6]

[1]Earth System Science Interdisciplinary Center, University of Maryland, College Park, MD, USA
[2]Global Modeling and Assimilation Office, National Aeronautics and Space Administration, Goddard Space Flight Center, Greenbelt, MD, USA
[3]Cooperative Institute for Research in the Atmosphere, Colorado State University, Fort Collins, CO, USA
[4]University of Leicester, Leicester, UK
[5]National Centre for Earth Observation, Leicester, UK
[6]Institute of Environmental Physics (IUP), University of Bremen FB1, Bremen, Germany
[7]now at Kaioa Analytics, Mundaka, Viscay, Spain
[8]LumenUs Scientific, Oklahoma City, OK, USA

**Correspondence:** Peter Somkuti (peter.somkuti@nasa.gov)

**Abstract.**

In this work, we present the results of an observing system simulation experiment (OSSE) in which we investigate the emergence of a surface reflectance-dependent bias in retrieved column-averaged dry-air mole fractions of methane (XCH$_4$). Our focus is on single-band type retrievals in the short-wave infrared (SWIR) at 2.3 µm. This particular bias manifests as artificial gradients in XCH$_4$ fields that relate to surface features on the ground and can, for example, cause erroneous estimates of methane source emission rates.

We find that even for near-ideal conditions (that being a perfectly calibrated instrument, perfect knowledge of meteorology and trace gas vertical distributions, and an absence of clouds and aerosols) a surface reflectance-related bias appears in the retrieved XCH$_4$. While the magnitude of the bias is much lower than is observed in e.g. real data from the TROPOspheric Monitoring Instrument (TROPOMI), the overall qualitative shape is strikingly similar. When we study a more realistic scenario by considering synthetic measurements that are affected by aerosols, the surface bias increases in magnitude roughly by a factor of 10. We hold all other properties of the synthetic measurements fixed, and thus can make the following statements about these surface biases from the 2.3 µm absorption band. First, the bias already appears in the near-perfect scenario, meaning that its origin is likely fundamental to XCH$_4$ retrievals from this particular absorption band, and using an optimal estimation-type retrieval approach. Second, the magnitude of the bias increases significantly when aerosols are encountered. As aerosols give rise to a magnification of the bias, we have implemented a retrieval configuration in which the retrieval algorithm knows the true aerosol abundance profiles along with their optical properties. With this configuration, the surface bias returns mostly to the level first seen when synthetic measurements were not affected by aerosols.

The results we present in this work should be considered for new missions where XCH$_4$ is a target quantity and the design relies on the 2.3 µm absorption band. Since the surface bias will likely emerge, it is crucial that a validation approach is planned

which sufficiently samples the needed range of surface reflectance in areas of near-uniform methane concentrations in order to capture the bias and thus correct for it.

## 1 Introduction

The wealth of data obtained by the TROPOspheric Monitoring Instrument (TROPOMI) very quickly revealed biases in the retrieved trace gases that were hidden until then, since no other trace gas instrument had both spatially dense coverage and footprint sizes on the order of a few square kilometers. Arguably, the most striking one is a bias that strongly correlates with surface reflectance features. Figure 1 shows one example over northeast Africa, in which the described bias is apparent; certain scenes with lower apparent surface reflectance, are accompanied by lower values of column-averaged dry-air methane mole fractions ($XCH_4$). This clear imprint of surface features onto the $XCH_4$ fields, driven here mostly by the contrast between rocky and sandy surfaces, is unphysical but routinely dealt with through a post-retrieval bias correction procedure (Hasekamp et al., 2019; Lorente et al., 2021; Schneising et al., 2023).

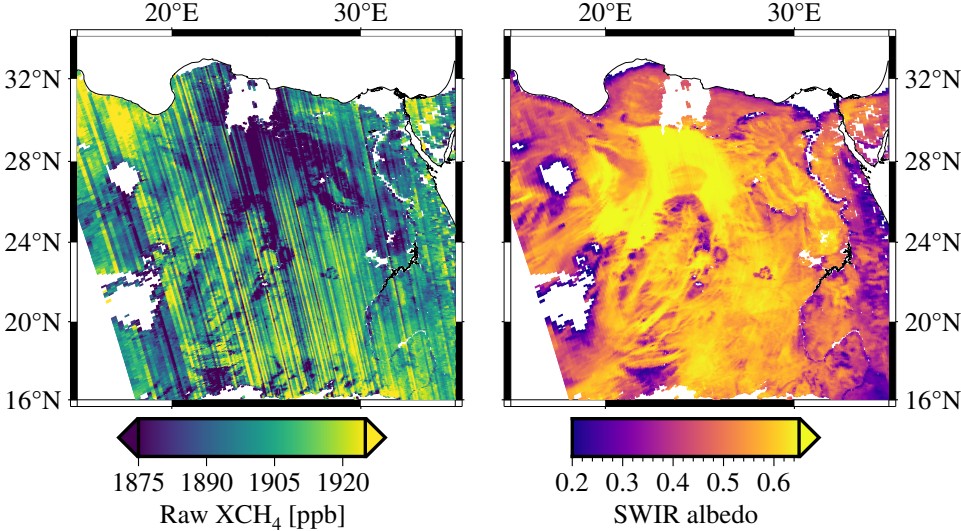

**Figure 1.** Surface reflectance bias example over northeast Africa. Shown are individual TROPOMI footprints, colored by either raw (without bias correction) retrieved $XCH_4$ (left) and apparent, retrieved surface albedo (right). Only scenes with albedo larger than 0.2 are shown here. Several surface features (right) are clearly visible to have corresponding gradients in retrieved $XCH_4$ (left). Colorbar ranges have been adjusted to exaggerate the effect by more strongly pronouncing the image contrast in both panels. No quality filters were applied, this figure is intended to show the raw retrieved methane column before any bias correction. This figure was produced from TROPOMI orbit 27865, processor version 2.4.0 (Copernicus Sentinel-5P, 2021).

Aerosols have been identified as a cause for systematic biases in retrievals since the early days of space-based missions that allowed for the estimation of greenhouse gas total columns. While not exclusively dedicated to greenhouse gases, the SCanning Imaging Absorption spectroMeter for Atmospheric CartograpHY (SCIAMACHY) instrument (Bovensmann et al., 1999), was

used to retrieve both $CO_2$ and $CH_4$ from the near-infrared (NIR) or short-wave infrared (SWIR) part of back-scattered spectra. Biases related to aerosols have been observed over the Sahara by Houweling et al. (2005) and were further studied by Aben et al. (2007) in a more comprehensive simulation exercise. Their conclusions are highly relevant to our study, as they also investigate a single-band retrieval configuration and observe the interaction between aerosol loading and surface reflectivity. We will contextualize our results in that regard in the Discussions section later on (Section 5). For the first dedicated $CO_2$

and $CH_4$ mission, the Greenhouse gases Observing SATellite (GOSAT) (Kuze et al., 2009), aerosols were also understood as a cause of bias in the retrieved total columns (Wunch et al., 2011; Uchino et al., 2012; Cogan et al., 2012). However, the way how aerosols interact with the various retrieved quantities is different for retrievals from GOSAT, compared to those of SCIAMACHY. GOSAT provides measurements of two separate absorption bands of $CO_2$, at 1.6 µm and 2.06 µm (in addition to thermal bands which are not relevant here), which provides effective de-coupling of the surface from the $CO_2$ concentration. As

such, significant surface-related biases have not been observed in GOSAT retrievals for $CO_2$. The Orbiting Carbon Observatory missions (Crisp et al., 2004; Crisp, 2015; Eldering et al., 2019) use roughly the same configuration in terms of observed spectral windows as the GOSAT mission, and retrieval algorithms used for either instrument are mostly interchangeable. Kulawik et al. (2019) found using another simulation study that the retrieved aerosol optical depth and retrieved surface albedos were indeed correlated when they inspected the posterior covariance matrices. In most related studies (e.g. O'Dell et al. (2018)), the major

drivers of biases are identified as retrieved surface pressure as well as the retrieved $CO_2$ profile shape. The retrieved aerosol optical depth and surface albedo contribute much less to the total bias correction (OCO-2 Science Team, 2023). It seems plausible that surface-aerosol interactions manifest as a different type of bias, for example through interference of surface pressure and aerosol optical depth retrieval. Regardless of the actual mechanism, the utilization of 3-band retrievals from GOSAT, OCO-2 and OCO-3 have made surface-aerosol biases less apparent, and the surface bias is no longer a dominant

contribution to the total observer errors.

As part of the algorithm development efforts for the GeoCarb mission (Polonsky et al., 2014; Moore III et al., 2018; Nivitanont et al., 2019; Somkuti et al., 2021; McGarragh et al., 2024), we investigated the emergence of a surface reflectance bias through a simulation study. We aim to answer questions related to this bias, namely: (1) whether we can reproduce the bias seen in TROPOMI retrievals through simulation-driven retrieval experiments, (2) whether we can determine what drives the

emergence of the bias, and, (3) if any mitigation strategy can be employed to reduce it.

The manuscript is structured in the following way. Section 2 describes the simulation set-up to produce synthetic observations. Section 3 then follows with the description of the retrieval algorithm used to derived the $XCH_4$ from the simulated radiances. Results are shown and discussed in Section 4, starting with retrievals from aerosol-free simulations (Section 4.1), and then moving on to one with realistic tropospheric aerosol abundances (Section 4.2), where the surface bias is first observed

in our study. In Section 4.3 we then augment the retrieval algorithm forward model by including the true aerosol information and analyze its impact on the surface bias. We summarize our results in Section 5 and discuss the both relevance of our study to the real-world biases seen in $XCH_4$ derived from TROPOMI measurements, as well as mention topics for future investigations.

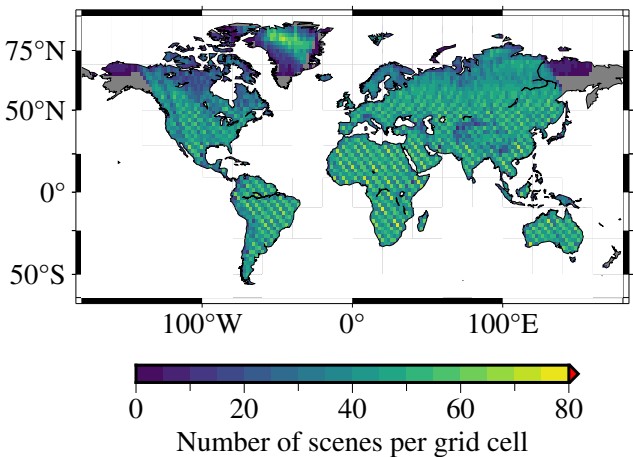

**Figure 2.** Spatial distribution density of the simulated scenes. Since the simulation locations are based on real OCO-2 locations, the sampling shows the expected striping pattern generated by the orbital movement of the spacecraft. Overall coverage is mostly the same, apart from eastern Siberia and Alaska where OCO-2 measures mostly in glint-mode, which we have excluded from our study. Note that for this and all other global-scale maps, features smaller than 10,000 km$^2$ in area are not drawn.

## 2 Simulation set-up

We use the same tested simulator framework, developed at Colorado State University (O'Brien et al., 2009; Polonsky et al., 2014) that has been successfully applied in other studies, such as Frankenberg et al. (2014); Eldering et al. (2019); Somkuti et al. (2021); McGarragh et al. (2024). Details on the inner workings of this orbit simulator can be found within these mentioned publications; we cover here only a short summary and focus on the aspects that are relevant to our study.

### 2.1 Sampling

Our full simulation set contains scenes derived from real OCO-2 (Crisp et al., 2004; Crisp, 2015) geo-location data within the period of January 2016 until March 2017. Due to the large amount of OCO-2 footprints, which is on the order of one million per day, the geo-location data was down-selected such that only one regular measurement every 10 seconds is retained, ignoring special measurement modes such as target mode. This corresponds to a scaling factor of 240, since OCO-2 measures 8 footprints roughly three times per second. At a global perspective, the general geographical coverage does not change with this down-selection and remains similar to that of the OCO-2 instrument. We retain only nadir-looking (down-looking at the sub-satellite point) scenes and drop any sun-glint following (pointed at the specular reflection of the direct solar beam) viewing modes. For this study, we only consider land surfaces and leave out Antarctica. The coverage of scenes is shown in Fig. 2.

As Fig. 2 illustrates, our set of scenes contains locations from all land masses with the exception of Antarctica. Since we are sampling MODIS BRDF coefficients (Schaaf and Wang, 2015) at every individual scene location, we obtain surface properties in the proper geographical context for each scene, as captured by the 500m-resolution MCD43A1 product. The surface for the

85 CH$_4$ band is spectrally flat since the MODIS instruments do not cover the shortwave-infrared region beyond $\approx 2.15$ µm. Thus, we take the BRDF coefficients from band 7 and use them for all wavelengths within the CH$_4$ window, without any spectral variation.

## 2.2 Clouds

We add cloud information from the International Satellite Cloud Climatology Project (ISCCP) by sampling the H-series dataset
(Young et al., 2018) at the scene locations and extract cloud flag, cloud type (liquid or ice), cloud path for optical density, and cloud top pressure for the vertical location of the cloud layer. Our measurement simulations fully account for clouds as part of the radiative transfer scheme. For the analysis of the post-retrieval quantities, however, we remove all scenes from the final analysis that contain any ice or liquid clouds. Our reason behind this choice is that we do not want to exercise cloud flagging algorithms for this work. We show in the following sections that tropospheric aerosols enhance the surface reflectance bias.
As such the scene sampling thus becomes representative of a mostly globally distributed set of locations, weighted by the probability of obtaining a cloud-free measurement at those locations and times.

## 2.3 Aerosols

As in Somkuti et al. (2021), we are utilizing reanalysis data ($0.75°$ spatial and 3-hourly temporal resolution) from the European Centre for Medium-range Weather Forecast's (ECMWF) Copernicus Atmosphere Monitoring System (CAMS) to assign
realistic aerosol abundance profiles to each scene (Bozzo et al., 2020). With the CAMS aerosol component, there are in total 11 aerosol mixtures: hydrophobic and hydrophilic organic matter, hydrophobic and hydrophilic black carbon, three sea salt mixtures, three mineral dust mixtures, and sulphate. The sea salt and mineral dust mixtures are separated into three spherical radius size bins each: $0.03 - 0.5$ µm, $0.5 - 5.0$ µm, $5.0 - 20.0$ µm for sea salt and $0.03 - 0.55$ µm, $0.55 - 0.9$ µm, $0.9 - 20.0$ µm for mineral dust. The sea salt, sulphate and hydrophilic organic matter mixtures are hygroscopic, meaning that the optical prop-
erties and the total aerosol particle counts for a given mass mixing ratio are dependent on the humidity. There is no humidity dependence for the mineral dust, hydrophobic organic matter and black carbon mixtures.

The process of integrating the CAMS model aerosol data into our orbit simulator and then the radiative transfer (RT) module is done as follows. In a pre-processing step, a library of aerosol mixture optical properties is generated according to the micro-physical parameters laid out in Appendix A2 of Bozzo et al. (2020). This step leverages a code for far-field scattering
calculations involving polydisperse mixtures of spherical particles based on Mishchenko et al. (2002). We calculate required optical properties (mass extinction coefficients, extinction cross sections, single-scatter albedo, phase function expansion coefficients) for each of the 11 mixtures at two wavelengths at both ends of the considered spectral range, and, if that mixture has humidity-dependence, for 12 different relative humidity values.

Then, vertically resolved aerosol (dry) mass mixing ratio profiles for distinct aerosol mixtures, as provided by CAMS, are
115 sampled at the specific locations and times for each scene. For each vertical layer $l$, the extinction optical depth $\tau_{a,l}$ for an

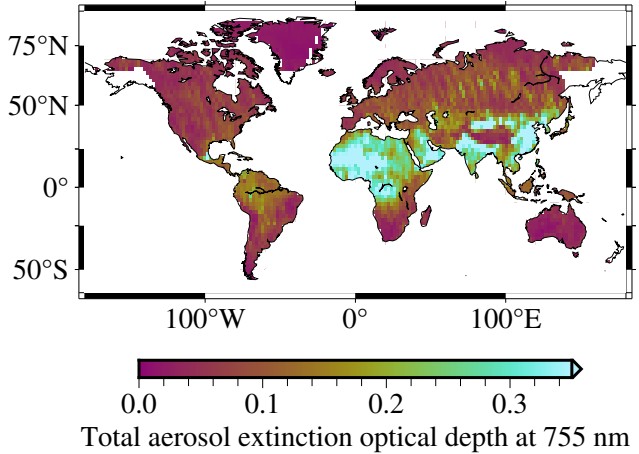

**Figure 3.** A map showing the geographical distribution of total-column aerosol extinction optical depth, gridded to $2° \times 2°$ grid cells.

aerosol mixture $a$ is given by

$$\tau_{a,l} = \alpha_{a,l,\rho} \cdot \text{MMR}_{a,l} \cdot \frac{\Delta p_l}{g_l}, \tag{1}$$

where $\alpha_{a,l}, \rho$ is the aerosol mass extinction coefficient for mixture $a$ ($[\text{m}^2 \text{ kg}^{-1}]$) at specific humidity $\rho$, $\text{MMR}_a$ is the mass mixing ratio ($[\text{kg kg}^{-1}]$) for mixture $a$, and finally $\Delta p_l$ is the pressure interval ($[\text{Pa}]$) for the given pressure layer $l$ and $g_l$ is the acceleration of gravity ($[\text{m s}^{-2}]$) at the center of the pressure interval. This extinction optical depth is calculated for each layer in the model atmosphere at the wavelengths at the edges of the wavelength window. Extinction and scattering profiles for each wavelength in between those edges are then interpolated through an Ångstrom exponent ansatz.

The radiative transfer scheme (Heidinger et al., 2006; O'Dell et al., 2006; Natraj and Spurr, 2007; O'Dell, 2010) finally ingests the total scene information, including the scattering properties for each mixture, to produce top-of-the-atmosphere (TOA) radiances, which are then fed into the instrument model which then results in a synthetic measurement. We do not apply instrument noise to the synthetic TOA radiances since we are interested in systematic errors. In the generation of the synthetic TOA radiances, we ultimately use high-accuracy calculations corresponding to 24 streams.

A pivotal aspect of our aerosol scheme is the complexity of the ingested aerosol information. Assuming there are contributions from all five hygroscopic (with 12 different humidity values) and all six non-hygroscopic aerosol mixtures, there is a total of 66 different aerosol components. The geographic distribution of total-column aerosol extinction is shown in Fig. 3.

We ingest the full aerosol profiles as prescribed by CAMS, rather than representing the vertical distribution as a simpler, parameterized shape, which is done in various retrieval algorithms (O'Dell et al., 2018; Lorente et al., 2021). Some examples of the vertical distribution of the aerosol mixtures is shown in Fig. 4.

The radiative transfer portion of the simulator can be run in a so-called "clear-sky" mode, in which absorption and scattering due to clouds and aerosols is ignored, resulting in a Rayleigh-only atmosphere. This mode allows us to produce two sets

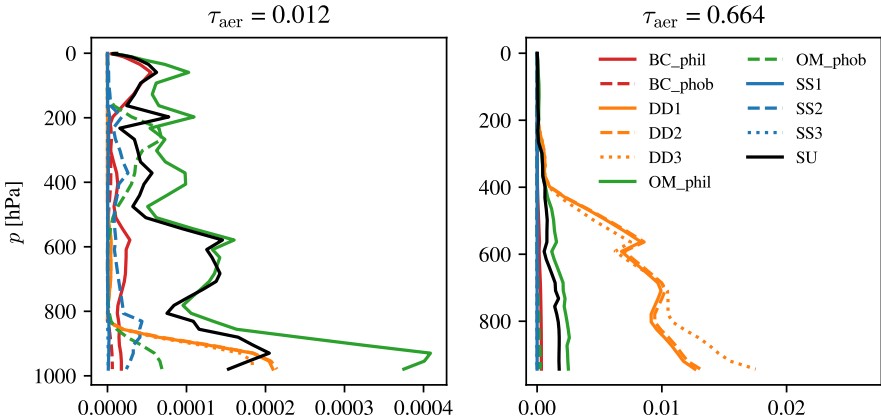

**Figure 4.** Two examples of the CAMS-derived aerosol profiles used in the generation of synthetic radiances. For this figure we aggregated all species into their respective type and size bins, regardless of their specific value of relative humidity. The example on the left shows a scene with low aerosol loading with contributions mostly from sulfates (SU) and hydrophilic organic matter (OM_phil). The example on the right, however, is dominated by all three size-bins of mineral dust (DD1, DD2, DD3). Hydrophobic organic matter (OM_phob), sea salt (SS1, SS2, SS3) and both types of black carbon (BC_phob, BC_phil) do not contribute significantly in these two examples. Note the complex shape of the vertical distributions in the left example, which would be very difficult to capture via a parametric description.

**Table 1.** Source and spatial resolution for the key datasets used in the simulations

| Data | Source | Spatial resolution |
| --- | --- | --- |
| Surface BDRF parameters | MODIS MCD43A1 (Schaaf and Wang, 2015) | $\approx 500$ m |
| $CO_2$, $CH_4$ and CO profiles | custom GEOS-5[†] (Molod et al., 2015) | $0.625° \times 0.50°$ |
| Aerosol profiles | CAMS reanalysis (Bozzo et al., 2020) | $0.75° \times 0.75°$ |
| Cloud parameters | ISSCP (Young et al., 2018) | $0.1° \times 0.1°$ |
| Meteorology | ECMWF ERA5 (Hersbach et al., 2020) | $\approx 31$ km |

[†] Trace gas profiles were sampled from a custom GEOS-5 run at 50 km spatial resolution, written out to $0.625° \times 0.50°$.

synthetic top-of-the-atmosphere measurements, one in which clouds and aerosols are present, and one without. Every other scene quantity is treated the same.

## 2.4 Data source summary

In Table 1 we summarize the various data sets that feed into our simulations and also note the spatial resolution at which those
data sets are provided.

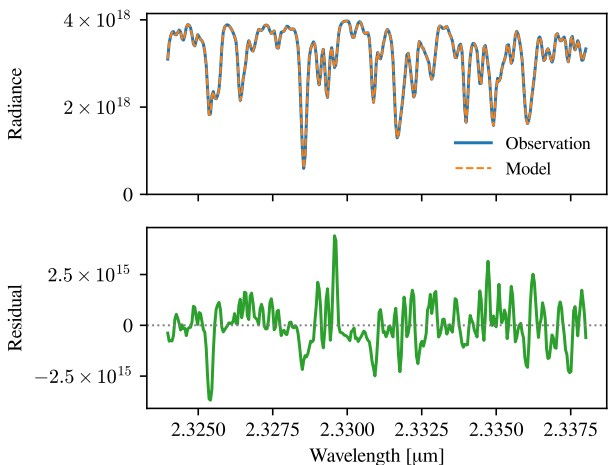

**Figure 5.** Illustrative example of the used retrieval window between 2.324 μm and 2.338 μm. The top panel shows both the synthetic TOA spectrum ("observation") along with the fitted spectrum produced by the retrieval algorithm ("model"), and the spectral residual is shown in the bottom panel. Both are in units of [ph s$^{-1}$ m$^{-2}$ sr$^{-1}$ μm$^{-1}$]. This particular example has a relative residual RMS of 0.03%.

## 3 Retrieval algorithm set-up

We use a single-band algorithm which has been developed for the GeoCarb mission and was demonstrated in an earlier study (Somkuti et al., 2021). The main retrieval window stretches from 2.324 μm to 2.338 μm, which contains absorption lines from CH$_4$, CO and H$_2$O. This retrieval window is similar to what Schneising et al. (2019) have used, and does not cover the entire
available range of the spectrometer. An example is shown in Fig. 5.

The forward model of the retrieval algorithm is conceptually equal to that of the simulator. Each scene is comprised of a layered atmosphere in which each layer is considered horizontally homogeneous in terms of their physical properties. At the layer boundaries, we set gas mixing ratios, pressure, temperature, and specific humidity. Optical properties of gases (CH$_4$, CO and H$_2$O) are calculated via pre-calculated spectroscopy tables derived from HITRAN2016 (Gordon et al., 2017) that are
150 sampled accordingly in the wavelength, temperature, pressure and humidity dimensions. More details on the calculation of those quantities can be found in e.g. Cogan et al. (2012); Wu et al. (2018); OCO-2 Science Team (2019).

In our retrieval algorithm, we can switch freely between two major modes to perform the radiative transfer calculations. The first one employs the non-scattering Beer-Lambert-Bouguer law, in which we only account for extinction from gases and Rayleigh scattering in an absorption-only atmosphere. The second mode invokes the XRTM radiative transfer library (Mc-
155 Garragh, 2020), which itself allows us to effortlessly switch between various numerical solvers, including different multiple-scattering ones. We use both radiative transfer modes as a means of understanding the robustness of our experiment. Here we want to emphasize that the non-scattering RT model will account for extinction due to Rayleigh scattering, as it is calculated as part of the layer-resolved total optical depths. Using the XRTM library, however, the retrieval algorithm forward model will include the contributions from Rayleigh scattering in addition to the extinction.

Within the forward model of the retrieval algorithm, we are generally free to choose an arbitrary vertical layering scheme, however for this exercise, we choose the same exact pressure layers (and layer boundaries) as the simulator forward model, 40 layers in total, in order to minimize the impact of simulation-retrieval mismatch. Additionally, we can ingest the same compound aerosol information as is used in the simulator forward model to obtain the same aerosol profiles and the corresponding scattering properties.

The inverse method is based on Rodgers (2000) and is an iterative Bayesian scheme that maximizes the *a posteriori* probability density function. Given an iteration $i$, the state vector for the next iteration $i + 1$ is calculated as

$$
\begin{aligned}
\mathbf{x}_{i+1} = \mathbf{x}_{\mathrm{a}} + \left(\mathbf{S}_{\mathrm{a}}^{-1} + \mathbf{K}^{\mathrm{T}}\mathbf{S}_{\varepsilon}^{-1}\mathbf{K}\right)^{-1}\mathbf{K}^{\mathrm{T}}\mathbf{S}_{\varepsilon}^{-1} \times \\
\left[\mathbf{y} - \mathbf{F}(\mathbf{x}_i) + \mathbf{K}\left(\mathbf{x}_i - \mathbf{x}_{\mathrm{a}}\right)\right],
\end{aligned}
\tag{2}
$$

where $\mathbf{x}_{\mathrm{a}}$ is the *a priori* state vector, $\mathbf{S}_{\mathrm{a}}$ is the associated *a priori* covariance matrix, $\mathbf{S}_{\varepsilon}$ is the diagonal instrument noise covariance matrix, and $\mathbf{K}$ is the forward model Jacobian matrix evaluated at iteration $i$. We mentioned earlier (Section 2) that the synthetic observations do not contain instrument noise, however we do use the GeoCarb noise model (Somkuti et al., 2021) for the calculation of a realistic $\mathbf{S}_{\varepsilon}$.

Our state vector contains the following elements: two polynomial coefficients to represent the spectrally varying Lambertian surface albedo, two polynomial coefficients to represent the assignment between spectral sample and wavelength (also referred to as dispersion), one scale factor for each of the considered trace gas profiles of $CH_4$, $CO$ and, $H_2O$, a temperature offset common to all vertical levels, and finally one value to adjust the spectral shift of the solar spectrum only. Values for the prior state vector $\mathbf{x}_{\mathrm{a}}$ are obtained as follows: gas scale factors are set to $1.0$, instrument dispersion coefficients are taken straight from the instrument model, the solar shift is set to $0.0\ \mu\mathrm{m}$, and the prior (and first guess) surface albedo is estimated from the radiances themselves via

$$
\rho_0 = \frac{\pi \cdot \max(I)}{\max(L_0) \cdot \cos\theta_0},
\tag{3}
$$

where $I$ is the measured TOA radiance, $L_0$ is the solar irradiance for the same retrieval window, and $\theta_0$ is the solar zenith angle. Note that we use the all points from the measurement that fall inside the retrieval band to calculate $\max(I)$. The prior value for the albedo slope coefficient ($\rho_1$) is $0.0\ \mu\mathrm{m}^{-1}$ for every scene. We make the choice to set the zeroth iteration to be equal to the prior state vector ($\mathbf{x}_0 = \mathbf{x}_{\mathrm{a}}$). As this is a so-called "scaling retrieval" in which the trace gas profiles are not changed within the iterative scheme, we must pick a profile shape to be scaled by the retrieval algorithm. We choose to use the true shape as they are used in the simulation forward model.

Iterations are halted as soon as one of these three criteria are met: the number of allowed iterations is reached, the change in reduced chi-squared statistic (modeled versus observed radiance) is smaller than $1\%$, the value in $d\sigma^2$ is less than the number of state vector elements, where

$$
d\sigma^2 = (\mathbf{x}_{i+1} - \mathbf{x}_i)\hat{\mathbf{S}}^{-1}(\mathbf{x}_{i+1} - \mathbf{x}_i),
\tag{4}
$$

with $\hat{\mathbf{S}}$ being the *a posteriori* covariance matrix defined as

$$
\hat{\mathbf{S}} = \left(\mathbf{S}_{\mathrm{a}}^{-1} + \mathbf{K}^{\mathrm{T}}\mathbf{S}_{\varepsilon}\mathbf{K}\right)^{-1}.
\tag{5}
$$

## 4 Results and analysis

The simulation experiments and subsequent analyses are organized in the following manner. First, we present a baseline scenario in which aerosols were ignored during the RT simulations and the retrieval forward model. Already in this baseline scenario we see a surface-dependent $XCH_4$ bias appearing. This is a key finding, as it establishes the fact that an interplay between apparent surface reflectance and retrieved $XCH_4$ is already present in an absorption-only atmosphere as a consequence of the retrieval forward model error. Then, we introduce aerosols into the RT simulations, and keep everything else exactly the same, i.e. not accounting for aerosols in the retrieval. This is where we observe a strong enhancement of the surface bias. Finally, we add the aerosol truth to the retrieval algorithm and observe a significant mitigation of the enhanced surface bias.

For the first two scenarios we use two different RT schemes - a non-scattering one, and a single-scatter model from the dedicated XRTM code. The non-scattering RT scheme is referred to as "non-sc" in various figures, and the single-scatter one is labeled as "SS". In a model atmosphere without scattering, the two approaches should yield the same result. However due to the numerical nature of RT codes, and the whole algorithm itself, small differences are to be expected and we utilize the two RT models as a form of validation. Ideally, we should observe biases and regional patterns thereof in the same places with both non-scattering and single-scattering models.

The experiments are laid out in a flowchart in Fig. 6 to allow the reader quick inspection of the relationship between simulator forward model set-up, and the corresponding retrieval algorithm set-up.

### 4.1 Baseline, clear-sky case

We first analyze the retrieval results based on simulations in which scattering from aerosols and clouds was ignored. This clear-sky case is considered the baseline scenario, labeled (CS1) and (CS2) in the flowchart shown in Fig. 6.

Here we like to remind the reader again that the forward model of the retrieval algorithm and the forward model of the simulator (which generates the synthetic measurements) are different. Since we force several aspects of the simulator and retrieval forward models to be the same, such as vertical layering, meteorological inputs, trace gas profile shapes and spectroscopy tables, this constitutes a *best-case* scenario. However we must emphasize that while many of the key ingredients in the simulator and retrieval forward models are the same, they do not produce numerically the same TOA radiances for the same set of atmospheric and surface properties. Thus, even for a clear-sky set-up for both simulator and retrieval forward model, there are forward model errors which cause retrieval errors.

Scenes from the clear-sky case are then run through the retrieval algorithm twice, with two different approaches for the radiative transfer calculations: once with the non-scattering model (non-sc), and once with the single-scattering model (SS). To remind the reader, Rayleigh scattering is present in the simulated radiances, as well as in the retrieval forward model - the non-scattering RT mode, however, does not produce contributions due to scattering at all. We chose to perform retrievals with the two mentioned RT schemes to provide robustness to the results.

For selecting the final subset of scenes to be analyzed, we apply very basic quality filtering criteria to remove retrievals that either did not converge within the maximally allowed number of iterations $N_{\mathrm{itermax}} = 3$, have a solar zenith angle $\theta_0$ above

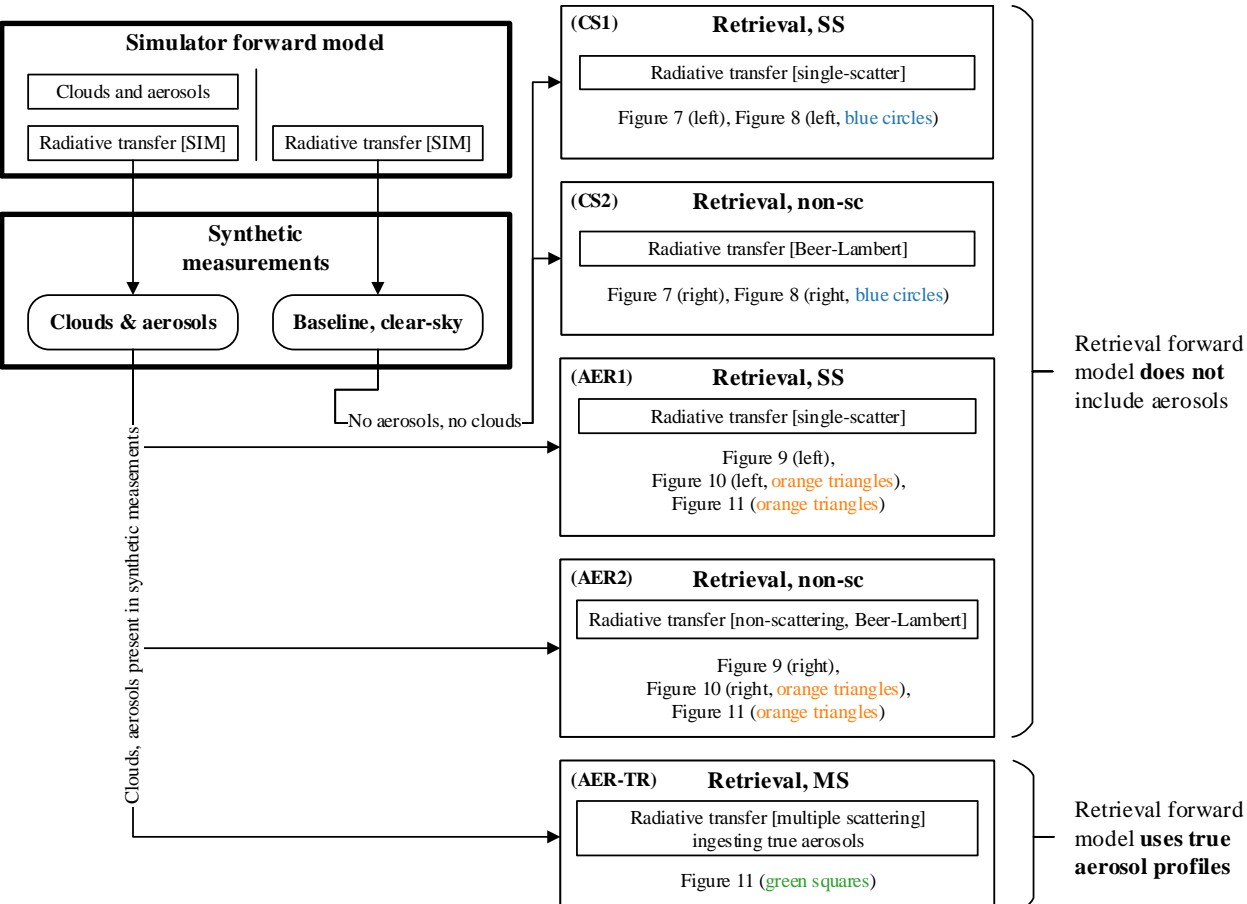

**Figure 6.** A flowchart illustrating the retrieval experiment set-ups. We generate two sets of synthetic measurements with the simulator forward model: one accounting for clouds and aerosols (left path) and one where clouds and aerosols are ignored (right path, clear-sky). The first two experiments enter both Fig. 7 and Fig. 10, labeled as (CS1) and (CS2): they represent the clear-sky retrievals from clear-sky simulations using two different RT model codes on the retrieval side. In bias plots (e.g. Fig. 10), they correspond to the blue circles. The second set of retrieval experiments, (AER1) and (AER2), follow the left path, where clouds and aerosols were present in the simulator forward model RT, however the retrieval RT still does not include aerosols. The results from (AER1) and (AER2) are shown in Fig. 9, Fig. 10 and Fig. 11, and they are always shown as orange triangles in the bias plots. Finally, the retrieval experiment denoted as (AER-TR) is based on the same synthetic measurements as (AER1) and (AER2), however the retrieval forward model now includes the true aerosol profiles, along with an appropriate multiple-scattering RT solver. In the final bias plot, Fig. 12, these results are shown as green squares.

the threshold of $75°$ or have a spectral residual reduced $\chi^2$ larger than $0.1$. Note that the $\chi^2$ values here are low due to the fact that we did not add instrument noise to the synthetic measurements, however we still use $\chi^2$ as a measure of fit quality.

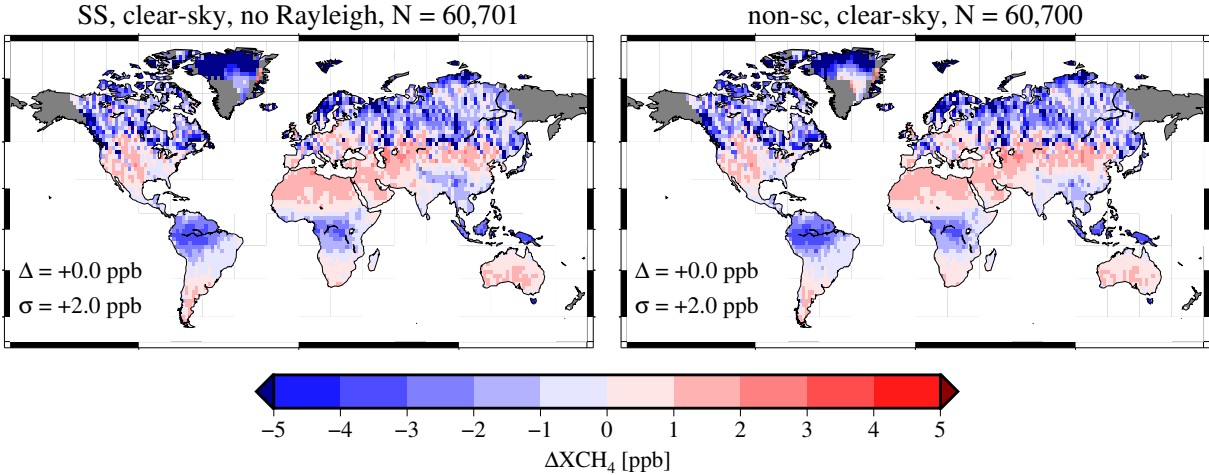

**Figure 7.** Maps of mean-removed retrieval errors for the clear-sky scenario in which no aerosols nor clouds were used during the forward radiative transfer simulations. Single-scattering radiative transfer on the left, absorption-only radiative transfer on the right - experiments (CS1) and (CS2), respectively (Fig. 6). The strongest regional highlights are seen in tropical forests where the surface reflectance at 2.4 μm is low. The total number of scenes for a given map is shown, since each set is quality-filtered separately, which can lead to a slightly different number of retrievals plotted.

As for bias correction, we only remove a single offset term which is the median of the ensemble difference between retrieved and true XCH$_4$. This correction brings the overall bias, per design, to 0.0 ppb, such that the error maps highlight regional-scale differences.

The maps in Fig. 7 show the XCH$_4$ errors for the two sets of retrievals (non-scattering, single scattering). Errors, meaning the difference between retrieved (and bias-corrected) and the truth, were calculated using the retrieval averaging kernels for each individual scene, as well as accounting for the prior methane profiles according to Wunch et al. (2011). Non-sc and SS configurations produce offsets of $-6$ ppb and $\leq 2$ ppb respectively, and small overall scatter. Errors show a geographic pattern with error enhancements in the tropics, and we find these errors be statistically significant, but weak linear functions of surface

reflectance and solar angles. The underlying cause for the difference between the results of the experiments (CS1) and (CS2) is the correct accounting for Rayleigh scattering in the retrieval forward model. The non-sc configuration does not produce scattered contributions to the TOA radiance, and thus shows larger errors overall, which can be observed by the scatter of the data in Fig. 7 (SS: $\sigma = 1.2$ ppb vs. non-sc: $\sigma = 2.0$ ppb). We have confirmed the impact of Rayleigh scattering by performing retrieval experiment (CS2) with a modified set-up in which the optical depth due to Rayleigh scattering was forced to be zero,

details on that modified experiment can be found in Appendix A.

The spatial distribution of errors shown in Fig. 7 provide some geographical context to the biases mentioned above. We see a contrast between areas with predominantly dark surfaces at 2.3 μm, such as the central African and South American tropical rainforests, and regions with much higher surface reflectance, such as the deserts. Note, however, that the overall magnitude

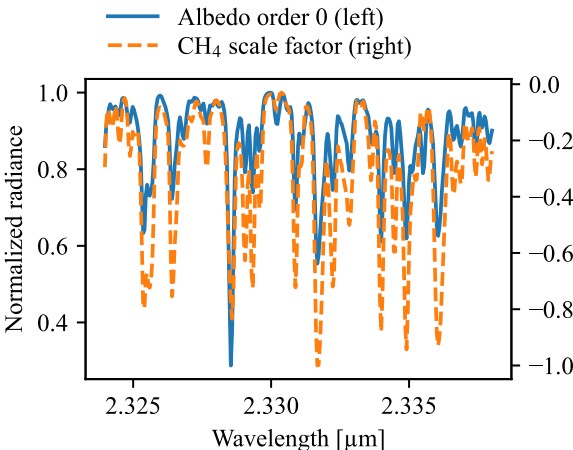

**Figure 8.** An illustration of the similarity between two Jacobians: Lambertian surface albedo polynomial order 0 (blue, solid, left axis) and CH$_4$ scale factor (orange, dashed, right axis). Both are shown normalized, but on separate ordinate axes to show the strong similarity of their shapes. The Pearson correlation coefficient of these two Jacobians is $R = 0.91$.

of these systematic errors is small. Noise driven errors for the GeoCarb instrument, for example, would be expected to be one
order of magnitude larger.

We assume the surface-dependent errors to emerge due to an inherent link between the retrieved XCH$_4$ and the apparent surface reflectance. This can be easily observed by analyzing the relevant entries in the Jacobian matrix of our forward model. Overlaying the Jacobians for the Lambertian surface albedo polynomial order 0 and the CH$_4$ profile scale factor, as we did in Fig. 8, we see that they match to some extent in their shapes, and their similarity can be stated with an overall correlation
coefficient of $R = 0.91$ for an example case with surface albedo of $\approx 0.1$. A more appropriate quantification of the similarity of those two state vector elements would be the construction of a correlation via the posterior covariance matrix $\hat{S}$: $C_{ij} = \hat{S}_{ij}/\sqrt{\hat{S}_{ii} \cdot \hat{S}_{jj}}$. This quantity $\mathbf{C}$ does not just represent the similarity of two entries of the Jacobian matrix $i$ and $j$, but also accounts for the instrument noise. Again, for this particular example displayed in Fig. 8, the relevant entry in $\mathbf{C}$ is approximately 0.51. The correlation is strong for this retrieval setup since the absorption features of methane in this wavelength range do not
show a distinct continuum. In plain terms, if such a correlation is seen in a retrieval forward model, the inversion will generally produce a weighted adjustment between the two offending state vector elements in order to minimize the cost function and minimize the mismatch between modeled and measured radiances. It is important to note that the strength of a correlation is not indicative of the magnitude of the effect on the retrieved quantities. So despite observing such a high correlation between retrieved surface reflectance and CH$_4$ scale factor, the overall impact is shown to be only a few parts per billion, or a few tenths
of a percent in relative terms.

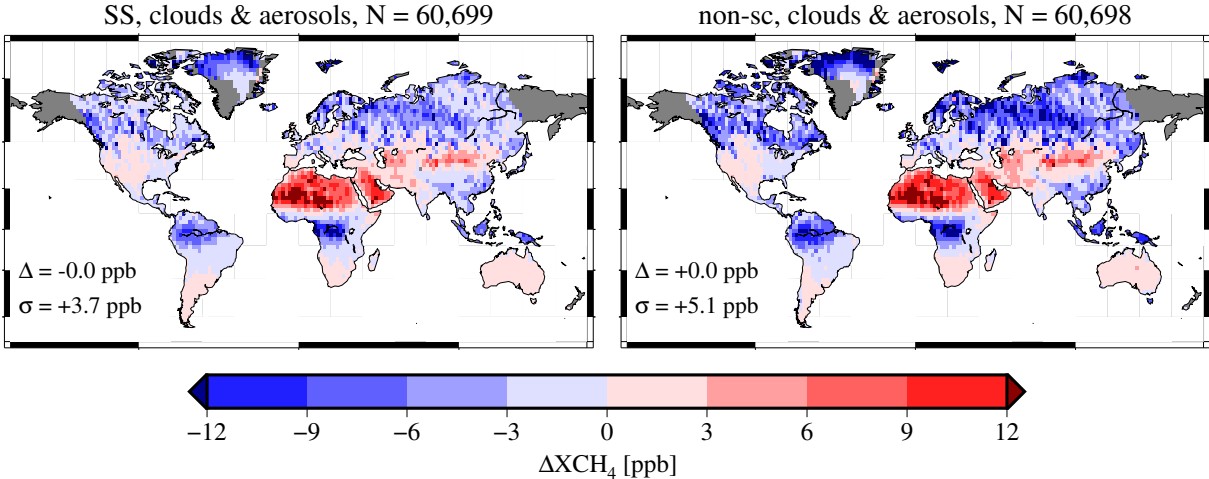

**Figure 9.** Error maps, similar to Fig. 7, however for model atmospheres in which aerosols and clouds are present. The regional contrasts between densely vegetated areas with low surface reflectance (tropical rainforests) and their surrounding areas (e.g. the Sahara desert) appear brighter in the 2.3 μm wavelength range. These are the results of expriments (AER1) and (AER2) when comparing to the flowchart in Fig. 6.

## 4.2 Inclusion of aerosols in the simulations

Building on the results presented in Section 4.1, we now introduce a single change. In the forward RT simulations, which produce the synthetic measured radiances, we switch on aerosols and clouds, however leave the retrieval set-up the same. In the flowchart (Fig. 6), these experiments are labeled as (AER1) and (AER2). The retrieval algorithms are ignorant to the fact

that the synthetic measurements now reflect a more realistic atmosphere in which multiple scattering via tropospheric aerosols has taken place. To remind readers, while the full produced data set includes scenes with thick water and ice clouds, we omit those scenes for this study.

We repeat the procedure from above and subtract the overall median error before producing the maps in Fig. 9. When compared to Fig. 7 (note the differently scaled colorbars), we observe a much stronger contrast between central tropical Africa

and the surrounding regions with brighter surfaces, and similarly see such a contrast between the scenes over the Tibetan plateau and surrounding areas. More importantly, the magnitude of the bias increased by a factor of roughly 4.

The change in the retrieved $XCH_4$ is purely driven by introducing aerosols into the forward model. We can represent this surface reflectance bias by grouping scenes into discrete bins of retrieved (or apparent) surface albedo and then calculating, for each scene, the ratio of true to retrieved $XCH_4$. This is shown in Fig. 10. For darker surfaces with apparent albedo less than

275 0.2, there is a clear low-bias, whereas a high-bias is observed for brighter surfaces with apparent albedo larger than 0.5. This observed bias is quantitatively comparable to that seen in TROPOMI-derived $XCH_4$, as first introduced in Lorente et al. (2021) and further elaborated in Lorente et al. (2023).

We make the following important observations. The surface reflectance bias already appears in clear-sky simulations and shows the underestimation of $XCH_4$ for dark surfaces in a very similar qualitative manner. Once the apparent surface albedo

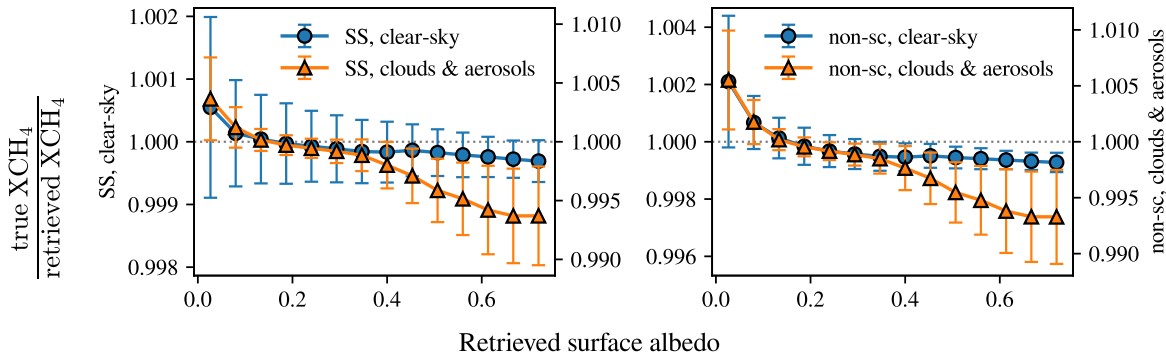

**Figure 10.** Curves that show surface reflectance bias: the ratio of true and retrieved $XCH_4$ as a function of the apparent surface albedo. For this figure, we first assign each retrieval scene to a bin according to the retrieved surface albedo. The circles (clear-sky) and triangles (with aerosols) then represent the median of all values within the bin, and the error bars are the robust standard deviation of the bin, calculated as the inter-quartile range divided by 1.349. This statistic is used for all other error bars in this manuscript. Note that the clear-sky observations (circles, left y-axis) and the ones with aerosols (triangles, right y-axis) are plotted on separate ordinates to make a qualitative comparison of the shape easier. The scale of the bias in the "cloudy & aerosols" scenario is roughly an order of magnitude larger. Blue circles represent the results of experiments (CS1) and (CS2), orange triangles represent the results of (AER1) and (AER2) when comparing to the flowchart in Fig. 6.

is larger than roughly 0.3, however, there is no significant bias seen for clear-sky scenes. The retrievals from the aerosol-laden scenes, however, show further dependence on the apparent surface albedo. Such a bias would imprint surface features on e.g. desert scenes like those shown in Fig. 1.

A phenomenological explanation for the shape of the bias curves in Fig. 10 has been stated in Aben et al. (2007). When aerosols are present, some fraction of the incident light is scattered into the field of view of the instrument. For scenes with low surface reflectivity, there is a relatively larger amount of light that has a shorter total light path from contributions which are scattered towards the instrument before reaching the surface. A retrieval algorithm that does not account for aerosols can thus only reduce the methane abundance to match the observed radiances. In Fig. 10, an underestimation of $XCH_4$ is equal to a ratio of true to retrieved $XCH_4$ larger than 1. On the other extreme, for very bright surfaces, the fraction of light that travels the full path through the atmosphere twice, is comparatively larger. In addition, contributions from multiple scattering due to tropospheric aerosols further increase the effective light path of photons. Without accounting for aerosols, the retrieval algorithm can only increase amount of $CH_4$ in the atmosphere to match the observed absorption lines, causing an overestimation. This overestimation shows up as a ratio of true to retrieved $XCH_4$ smaller than 1. We further note that this explanation should hold true for an aerosol-free atmosphere in which Rayleigh scattering occurs. Despite the fact that the total-column optical depth due to Rayleigh scattering at 2.3 µm amounts to only $\approx 10^{-4}$ (Tomasi et al., 2005), the impact is significant and can be observed in these bias curves (see also Appendix A).

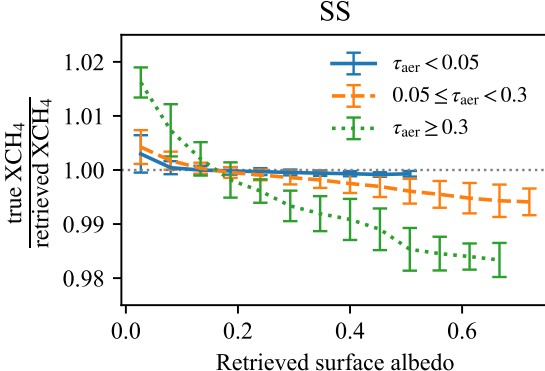

**Figure 11.** Bias curves, similar to Fig. 10, but without clear-sky simulations. The results of experiment (AER1) are shown. Retrieval results are split into sub-sets of different regimes of total aerosol extinction optical depth $\tau_{aer}$ (at a reference wavelength of 755 nm). This figure shows that the underestimation of $XCH_4$ for dark surfaces with albedo less than 0.1 is mainly driven by scenes with larger aerosol loadings.

We note that the surface reflectance bias discussed in Lorente et al. (2021, 2023) is larger in magnitude, but shows the same general shape as our result in qualitative terms. The observed surface reflectance bias as shown in Fig. 10 is the result of a global aggregate. When the same figure is produced for various subsets, separately, however, we see that the strength of the bias changes as a result of the amount of aerosols within that subset. In Fig. 11, we group the global set of scenes into two subsets of different total aerosol extinction optical depths $\tau_{aer}$. Through this figure, we can observe that the underestimation of $XCH_4$ for darker surfaces gets larger with increasing aerosol loadings. Notice the kink near albedo value of $\approx 0.4$ in Fig. 10 that is seen for the simulation set that includes clouds and aerosols (orange, triangles), which is absent in any of the curves in Fig. 11. Since the total aerosol extinction per scene is not equally distributed amongst the bins of apparent surface reflectance, we suspect the observed kink in the bias curve to be a result of a sampling bias which blends together the various curves of different $\tau_{aer}$ regimes. This is investigated in Fig. 11 in which we produce bias curves for different bins of aerosol loadings.

We do not find any significant impact of the aerosol single scattering albedo on the bias, suggesting that the total aerosol extinction is the main driver in the case of single-band retrievals of this type.

### 4.3 Mitigation by accounting for true aerosol profiles in the retrieval

In Sections 4.1 and 4.2 we have observed that the surface reflectance bias is already present in clear-sky conditions, but is strongly enhanced when aerosols are introduced in the RT simulations that produce the synthetic measurements. Consistent with that notion is the fact the dependency of the error grows with larger aerosol abundances, as shown in Fig 11.

An obvious way to mitigate the surface reflectance bias is to inform the retrieval algorithm about the aerosol scattering profile that is present in the scene. We implement this straightforwardly by adding the layer-resolved aerosol extinction and scattering optical depths to the (forward) model used during the retrieval, appropriately adding the phase function expansion coefficients

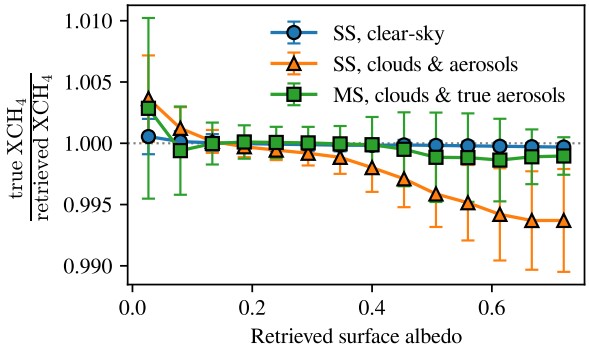

**Figure 12.** Surface reflectance bias similar to Fig. 10. In this figure, however, the third curve (squares, green), is derived from retrievals for which the true aerosol profiles were ingested as part of the retrieval algorithm forward model. This is expriment (AER-TR) in the flowchart (Fig. 6). We observe that, when comparing to the unmitigated runs (triangles, orange) of experiment (AER1) no significant bias remains. This suggests that better constraining aerosols is key to mitigating these types of retrieval biases.

with their correct relative weights to obtain a match of the total optical properties used by the simulator (see Sec. 2). Then, we also switch the RT model in the retrieval algorithm to use multiple scattering via a discrete ordinate solver with 16 streams (8 per hemisphere). This is necessary since using a single-scattering RT model, while also incorporating the true aerosol profiles in the retrieval forward model, does not result in a mitigation of the observed bias. In the flowchart (Fig. 6), this set-up is labeled as experiment (AER-TR).

The result is shown in Fig. 12, in which we overlay the described approach with an earlier result that used a single-scattering RT model without any knowledge of the aerosol profiles. For nadir viewing geometry, the bias curve exhibits much smaller dependency on the surface albedo when compared to the original approach. This result shows that the mitigation strategy is successful in reducing the surface bias and almost brings it to the same level as observed for the clear-sky scenario.

## 5    Discussion & Conclusions

In this study, we analyzed the impact of tropospheric aerosols on biases of $XCH_4$ obtained from single-band retrievals from the 2.3 μm absorption window. We were able to demonstrate that a weak surface-dependent bias is present already in clear-sky conditions, however aerosols can amplify those retrieval biases, and the effect grows with aerosol abundance as shown in Fig. 10.

     The significance of our result is related to actual findings from the TROPOMI instrument, which have been discussed by
Lorente et al. (2021, 2023). Surface reflectance biases in retrieved $XCH_4$ are a troublesome feature, since surface patterns on the ground will manifest as gradients of total column methane which can lead to wrong estimates of e.g. emission rates or the emergence of artificial features (Froitzheim et al., 2021; Schneising et al., 2023). In the past, studies have required an ad-hoc

correction to remove the surface reflectance bias in the $XCH_4$ fields (Liu et al., 2021) or remove scenes entirely which show a large correlation between $XCH_4$ and surface albedo (Sadavarte et al., 2021).

While the bias that we observe in our study is qualitatively similar to that seen in Lorente et al. (2021, 2023), we want to highlight that there are several differences in our instrument model and that of the TROPOMI instrument, as well as some key differences in our retrieval approach. First, the spectral resolution of the TROPOMI spectrometer for the SWIR band is $\approx 0.25$ nm, where as our instrument model, derived for the GeoCarb instrument, is closer to $\approx 0.12$ nm. Further, we utilize a single-band retrieval, whereas Lorente et al. (2021, 2023) co-retrieve the oxygen A-band at 0.76 µm, which, in general, should

allow for better constraining of the retrieved aerosol abundance. Lastly, in our simulations (Section 2), we do not introduce any instrument or calibration artefacts such as, but not limited to, remaining stray light, imperfect radiometric calibration or imperfect knowledge of detector (non-)linearity. This, in turn, also supports the argument that the biases observed in TROPOMI data are not caused by any instrument-related issues or calibration deficiencies, but are intrinsic to the retrieval approach from the 2.3 µm band.

In Lorente et al. (2023), the bias takes on a slightly different shape when the spectral dependence of the retrieved Lambertian surface albedo is changed from a second- to a third-order polynomial. This is not a feature that we can investigate with our simulations since the surface model in our simulations, which produce the synthetic observations, is spectrally flat. This is solely a constraint of the underlying observation-based dataset (Schaaf and Wang, 2015) which does not provide measurements beyond $\approx 2.15$ µm. Therefore, we also cannot investigate the impact of adjusting the order of the retrieved Lambertian surface

albedo polynomial.

    These types of aerosol-driven biases in retrieved trace gas columns have been studied in the past and are not exclusive to TROPOMI. We return to the notable studies of high relevance to ours by Houweling et al. (2005) and Aben et al. (2007), which both explored the topic in the context of the SCIAMACHY instrument. In Houweling et al. (2005), they find significant $XCO_2$ biases in the Sahara region due to high aerosol loadings, which Aben et al. (2007) further elaborated on via a sensitivity study

with retrievals from simulated observations. By re-ordering our results in a different manner in Fig. 13, we can reproduce their findings qualitatively, specifically Fig. 4 in Houweling et al. (2005) and Fig. 3 in Aben et al. (2007). One can consider our results to be an extension of their studies to $XCH_4$ in the 2.3 µm absorption band (rather than $XCO_2$ from the 1.6 µm band). The stark difference in magnitude of the effect is likely due to the instrument characteristics. For example, SCIAMACHY's spectral resolution ($\approx 1.35$ nm) is over five times lower than that of TROPOMI ($\approx 0.25$ nm). The deciding common aspects

of our study and those of Aben et al. (2007) are the following: (1) both studies use a single spectral window to retrieve a trace gas, (2) the absorption features within the chosen retrieval window are such that there is no clear continuum to sufficiently de-couple surface reflectance from gas concentration, (3) the studied atmospheric states contain weakly scattering aerosols up to total optical depths of $\approx 1$. Our study does not, however, include thin high-altitude cirrus clouds.

    In Section 4.3, we demonstrate the impact of perfect knowledge of the tropospheric aerosol profiles. Once the retrieval

algorithm is aided by inserting the true aerosol distributions into each scene, most of the surface reflectance bias is mitigated, as shown in Fig. 12, where we obtain results that are similar to those for the clear-sky scenario. This result shows the importance of better constraining the overall aerosol information for use in retrieval algorithms, as has been previously stated in different

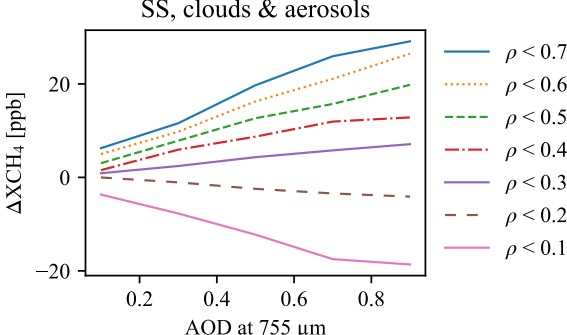

**Figure 13.** A re-ordered view of the results from Fig. 10. Each curve represents a sub-set of scenes whose retrieved, apparent surface albedo ($\rho$) falls into some bin. This way of ordering the results mimics Fig. 3 from Aben et al. (2007) and shows a qualitative match. The order of the legend items is the same as they appear in the figure: the lowest curve (pink, solid) contains scenes with $\rho < 0.1$, the curve above (brown, dashed) contains scenes with $0.1 < \rho < 0.2$, and so forth. Further, the quantity plotted is not the ratio of true over retrieved $XCH_4$, but the difference of retrieved minus true $XCH_4$.

contexts (Bell et al., 2023; Rusli et al., 2021). Aerosol-driven biases are of such concern, that for the upcoming Copernicus CO2M mission, developed by the European Space Agency to monitor anthropogenic carbon dioxide emissions, a dedicated aerosol instrument will be part of the payload in order to improve the quality of the $XCO_2$ retrievals (Sierk et al., 2021). While our study does not allow for any conclusions to be drawn for the CO2M mission regarding possible surface biases and their enhancement due to aerosols, the specific instrument could be investigated using our observing system simulation experiment set-up.

We have shown that incorporating the true aerosol information mostly removes the surface reflectance bias for nadir-viewing observations, however implementing this approach in a real science data processing scenario might not be feasible. It needs to be shown yet if globally covering aerosol forecasts, e.g. CAMS (Copernicus) or GEOS-5 (Molod et al., 2015), are close enough to the truth to be treated as such in the retrieval forward model for the purpose of mitigating the discussed bias. In general, limitations on data processing resources might ultimately necessitate the usage of faster forward models that could introduce biases similar as shown here. We have observed that the bias seems to be driven by the total aerosol extinction optical depths in the scene, rather than the vertical distribution or how absorbing the aerosols are. A future study could examine whether the a simpler aerosol profile than those shown in Fig. 4, while conserving the total aerosol extinction optical depth, would be just as effective in mitigating the surface reflectance bias.

Another important aspect of our study is the chosen absorption window. We investigated the 2.3 µm window as both the TROPOMI and the GeoCarb instruments are equipped with corresponding spectrometers.

One can expect that the same behavior arises with retrievals from the 1.65 µm window since the fundamental mechanism that drives the bias is the same, despite the spectroscopic features having different characteristics. The often-used "proxy method"

(Frankenberg et al., 2005) provides a solution for some spectral regions, in which a ratio to some reference trace gas is retrieved, thus canceling out any biases from unaccounted light path modification due to aerosols. This is the retrieval strategy of choice for the MethaneSAT (Chan Miller et al., 2023) mission. In fact, Chan Miller et al. (2023) present first results from the airborne MethaneAir instrument and do not observe any strong surface-related biases.

For point source-related applications, such as emission rate quantification, our results remain transferable only in a limited sense. In a clear-sky environment, the surface variation will already imprint onto the retrieved XCH$_4$ field, which can impact the estimation of emission rates. This has been studied to some extent by Jongaramrungruang et al. (2021) for various instrument configurations, focused on spectral windows much wider than that used in our study, and more realistic surfaces. They find, in general, lower precision errors when retrieving from the 2.3 μm band, compared to the 1.6 μm one, however they do not account for scattering from either Rayleigh scattering or aerosols. In scenarios with substantial background aerosols, the surface imprint onto the retrieved XCH$_4$ would show higher magnitude (compared to the same scene on a day without aerosols). A more difficult scenario would be the co-emission of aerosols from the methane point source, as an empirical correction using retrievals outside of the main plume might not fully capture the bias inside the plume.

In our analysis we found scenes measured in sunglint-following viewing geometry to behave distinctly different from nadir-viewing ones. The surface bias shows a qualitatively different shape and the mitigation effort through implementing aerosol truth information did not work as well as with nadir-viewing scenes. Efforts to understand the cause of this discrepancy did not yield any satisfying answers. Given the small absolute magnitude of the effect, however, we hypothesize that some inconsistency in the set-up of the retrieval RT and the simulation RT codes is the main cause.

Finally, we want to again highlight the results of the clear-sky baseline scenario presented in Section 4.1. Even in almost ideal circumstances where meteorology, spectroscopy and trace gas profiles are known perfectly, an optimal-estimation based retrieval exhibits a small but significant surface-dependent XCH$_4$ bias. We suspect that this is an inherent consequence of the 2.3 μm band, which does not have a clear continuum portion via which surface reflectance and methane abundance can be sufficiently disentangled. Thus, a mission designed for the remote sensing of methane from the 2.3 μm absorption band will likely require a surface bias correction procedure as a core part of its operations concept. As long as the surface bias is sufficiently characterized, an appropriate correction can effectively mitigate the impact on the retrieved XCH$_4$ field.

*Code and data availability.* The results of the study, i.e. the results of retrievals and the truth values, as well as a Python notebook that produces the figures used in this manuscript, can be downloaded from Zenodo at https://zenodo.org/records/13285730 (DOI:10.5281/zenodo.13285730).

## Appendix A: The impact of Rayleigh scattering on the clear-sky experiments

In this appendix section we are clarifying the impact of Rayleigh scattering on the clear-sky experiments, as mentioned in Section 4.1.

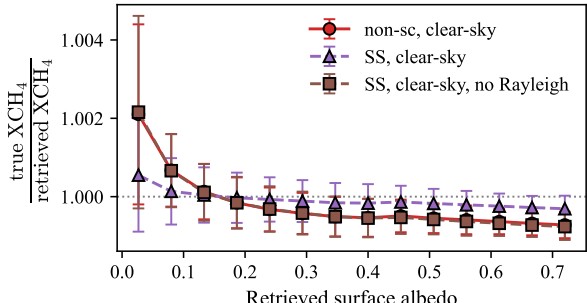

**Figure A1.** This figure (similar to Fig. 10) shows the impact of Rayleigh scattering on the surface bias. The clear-sky experiments (CS1) and (CS2) are shown in red (circle markers) and purple (triangle markers). The modified experiment is shown in brown (square markers), which almost fully overlap the round markers of experiment (CS1), therefore showing the impact of Rayleigh scattering on the surface bias. Note the difference in colors and markers when comparing to Fig. 10.

Observing again Fig. 10, we notice that the magnitude of the bias seen in the two experiments (CS1) and (CS2) (see Fig. 6 for the meaning of those labels) is different. The only change between experiments (CS1) and (CS2) is the used RT scheme in the retrieval: (CS1) uses the Beer-Lambert-Bouguer law for an absorption-only atmosphere, (CS2) uses the single-scattering solver through the XRTM library (McGarragh, 2020). In both experiments, the retrieval forward model computes all necessary contributions from Rayleigh scattering and produces the same total optical parameters: optical depth, single-scatter albedo, and the appropriate phase function. In experiment (CS1), however, only the optical depth is passed on to the RT routine which computes the TOA radiances. Rayleigh scattering contributions are always present in the synthetic observations (the simulation forward model).

It can be shown that the difference in magnitude of the bias can be fully attributed to Rayleigh scattering. We perform a control experiment, which is a modified run of experiment (CS2). We then manually set the optical depth due to Rayleigh scattering to zero everywhere, which automatically leads to the single-scatter albedos being zero everywhere as well. Therefore, we have a retrieval forward model that is does not account for Rayleigh scattering at all.

After performing the same actions on the resulting dataset as before (basic quality screening and bias correction), we compare the results of this modified run with the results of experiments (CS1) and (CS2). In Fig. A1, we can now observe that the modified run is near-identical to experiment (CS1). We note that this is a surprising result, as Rayleigh scattering often is ignored in studies when wavelengths $> 1\,\mu m$ are concerned (e.g. Jongaramrungruang et al. (2021)). Our control experiment shows that the inclusion of Rayleigh scattering can make a significant difference and warrants consideration.

*Author contributions.* PS devised the study, developed the retrieval algorithm, wrote the analysis and drafted the first revision of the manuscript. GM, CO, LO, SC and PS produced the simulation dataset. ADN, LV and HB implemented the CAMS aerosol scheme. All edited the manuscript into its final submitted form.

*Competing interests.* We declare no competing interests.

*Acknowledgements.* PS, GM, CO and SC were funded by NASA through the GeoCarb Mission under award 80LARC17C0001. Work at Colorado State University was supported via subcontract 2017-40 with the University of Oklahoma for the GeoCarb mission. Additionally, PS, GM, CO, LO and SC received support through NASA's Carbon Monitoring System (CMS) program under award NNH20DA001N-CMS 20-CMS20-0011. PS was also funded through NASA's Cooperative Earth Systems Science Research Agreement (CESSRA) with grant number 80NSSC23M0011.

Our study made use of following tools for data processing and analysis: the Numpy array programming tools (Harris et al., 2020), SciPy (Virtanen et al., 2020), Statsmodels (Seabold and Perktold, 2010), pandas (McKinney et al., 2011) and Jupyter (Kluyver et al., 2016). Visualizations were produced with Matplotlib (Hunter, 2007) and PyGMT (Uieda et al., 2023), which leverages GMT6 (Wessel et al., 2019).

Finally, we acknowledge the exceedingly helpful comments of two anonymous referees who helped us finalize the manuscript and pointed us towards understanding the impact of Rayleigh scattering in our retrival experiments.

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
