# Peer review of "Surface reflectance biases in $XCH_4$ retrievals from the 2.3 $\mu$ m band are enhanced in the presence of aerosols"

_Atmospheric Measurement Techniques, 2024_

## Author Comment (AC1)

**Final Response for AMT-2024-145**

Peter Somkuti1,2, Gregory McGarragh3, Christopher O'Dell3, Antonio Di Noia4,5,6, Leif Vogel4,5,7, Sean Crowell8, Lesley Ott2, and Hartmut Bösch4,5,6 1Earth System Science Interdisciplinary Center, University of Maryland, College Park, MD, USA 2Global Modeling and Assimilation Office, National Aeronautics and Space Administration, Goddard Space Flight Center, Greenbelt, MD, USA 3Cooperative Institute for Research in the Atmosphere, Colorado State University, Fort Collins, CO, USA 4University of Leicester, Leicester, UK 5National Centre for Earth Observation, Leicester, UK 6Institute of Environmental Physics (IUP), University of Bremen FB1, Bremen, Germany 7now at Kaioa Analytics, Mundaka, Viscay, Spain 8LumenUs Scientific, Oklahoma City, OK, USA

Correspondence: Peter Somkuti (peter.somkuti@nasa.gov)

Below are the point-by-point replies to review comments RC1 and RC2 for submission AMT-2024-145. We separated the two sets of comments into sections 1 (RC1) and 2 (RC2), and subsections within that address each comment individually. For each subsection, the original review comment is stated in **bold** text at the beginning of the subsection, followed immediately by our reply. If changes to the manuscript were made in the form of added text, they are put inside a distinct frame such as this:

Example text added to the manuscript.

- 5 New figures or tables are generally added right below the blue frame that surrounds newly added text. Deleted text passages will not be explicitly displayed, but will be seen in the difference document, uploaded later on. Figure numbers are generally organized as follows: in text passages newly added to the document (blue-shaded frames), the figure number represents the number in the revised manuscript; in replies, the figure number is usually followed by e.g. "in the original manuscript" to clarify that the figure number represents the number as it appears in the initial submission; some figures inside this document 10 are being referenced, and those are specifically named "RFIG" (and clickable in this PDF).
  - 1 Review 1 RC1

**1.1 Comment 1**

**Clarify how BRDF wavelength dependence is treated in the OSSE forward simulation, as it is a key difference between forward and inverse models.**

15 Indeed, in our forward model simulations, the BRDF wavelength dependence is based on a linear interpolation of BRDF parameters from the MODIS MCD43A1 dataset, which is spectrally given for the first 7 MODIS bands. Since the  $CH_4$  band

in our simulations and retrievals at 2.3  $\mu$ m is outside of the wavelength range spanned by those bands, we use the values from band 7 (roughly 2.15  $\mu$ m) across the band, without any spectral dependence. Therefore, in the forward simulations, the surface reflectance is spectrally flat. We added text at the end of section 2.1 to clarify this point, as well as make corresponding notes in the discussion section. Please also note comment 2.2 that further discusses this matter.

The surface for the CH4 band is spectrally flat since the MODIS instruments do not cover the shortwave-infrared region beyond  $\approx 2.15 \,\mu\text{m}$ . Thus, we take the BRDF coefficients from band 7 and use them for all wavelengths within the CH4 window, without any spectral variation.

**1.2 Comment 2**

Do ISCCP cloud observations cover the simulated period? When combined with the CAMS there could be inconsistencies from where the chemical transport model simulates clouds. For instances where there are clouds in CAMS but not ISCCP, the AOD may be overestimated because hygroscopic growth is being accounted for.

25 We use CAMS reanalysis for the forward simulation; as stated in Rémy et al. (2019), the CAMS system assimilates AOD from MODIS collection 6, and the Polar Multi-Angle Product, which itself is based on observations from GOME-2, AVHRR and IASI. The observations in the ISCCP dataset, as can be read in Young et al. (2018), themselves are a composite of various polar-orbiting and geostationary platforms that are aggregated into a combined dataset.

We have sampled both the ISCCP cloud dataset and the CAMS aerosol reanalysis at the same time/location pairs for each
scene - so they indeed should be compatible in the sense that any biases occurring in the reanalysis due to extended cloud cover would impact scenes that likely contain clouds in our simulations, as informed by ISCCP.

We have not, however, gone through an exercise to validate any aspects that could relate to a potential mismatch between the ISCCP cloud flags (or other parameters we utilize, such as cloud top height) and errors in the CAMS reanalysis. In our study, we are primarily concerned with producing a realistic and geo-spatially reasonable atmospheric state that is informed by both

35 models and observations. We then reference our retrieved results against this truth. We agree that it could be possible that the overall distribution of clear-sky aerosol loadings are skewed with respect to reality, but we believe that this would have negligible impact on our results.

**1.3 Comment 3**

Earlier in the section it was stated that Rayleigh scattering was insignificant, but is this still true for the small magnitude of the changes being considered in Fig. 7? For instance the negative bias over the dark surfaces seems consistent with atmospheric scattering, since photon paths from light scattered from the atmosphere will have a greater contribution to the total radiance at the sensor relative to the brighter surfaces. To first order this would be radiation from the solar beam directly scattered into the path of the sensor, which would effectively shorten the light path, making a forward model that does not account for it reduce the CH4 column, as is shown in the figure. Since the magnitude of the biases

RFIG 1. Fig. 9 in the originally submitted manuscript.

45 reduces between the non-sc and SS cases, this supports Rayleigh being potentially important for biases at this level. It should also be noted that a 2-5 ppb bias shown in the Figure may actually be significant depending on the application. E.g. A lot of diffuse agricultural sources produce enhancements around this magnitude. If anything the results show how close to perfect a retrieval would have to be to quantify these.

We thank the reviewer for this highly observant comment, they have indeed pointed out an aspect of our simulations and

50 retrievals that we failed to observe, or rather, failed to realize the significance of accounting for Rayleigh scattering at 2.3 μm. In order to assess the importance of accounting for Rayleigh scattering, we have performed the retrievals again in several different configurations and are focusing on the impact it has on the surface bias.

First, we re-visit the originally offending data that corresponds to the two experiments named (CS1) and (CS2) - retrievals done on clear-sky simulations with the single-scatter RT module (CS1) and the non-scattering Beer-Lambert type RT (CS2).

55 Fig. 9 in the original manuscript does not show those two experiments on the same panel, they are the blue curves with round markers on both panels in RFIG. 1.

We can overlay the two curves representing (CS1) and (CS2) into a single figure, shown here in RFIG. 2. One can clearly see the difference between both curves that represent this surface bias: the experiment (CS1) with the non-scattering RT exhibits a larger magnitude, although the overall structure is the same.

- 60 Initially, we assume this discrepancy to be the result of minute details related to the different RT modules. After all, the RT module used for (CS2) is a fully-fledged RT framework capable of utilizing multiple solvers in both scalar and vector mode. For example, our retrieval algorithm, when using that XRTM module, enables the pseudo-spherical approximation by default and even though we do not tend to be in the regime of viewing and solar angles where that would be a major driver for errors, one could imagine small discrepancies become evident in an analysis such as the one we are performing this was in fact the
- 65 reason why we used both RT models: to make sure that our results are robust.

RFIG 2. Overlaying the two curves (blue, round markers) from both panels from RFIG. 1. Note the difference in colors.

---

## Referee Report (RR1)

**Response to Comment 2:**

My point was just that for cases where the met model used by CAMS is simulating a cloud where ISCCP is flagging as clear, then the RH in the simulation will be ~100%, which will yield an unphysically high AOD due to hygroscopic growth. Provided the model and ISCCP differences are small this won't matter too much, but I would prefer just using the cloud data from the ERA5 model to be self consistent in the OSSE. As far as I can tell the main reason for using ISCCP is to simulate realistic observability statistics. My intuition would be to opt for self-consistency above this. It is a relatively minor point (not worth redoing simulations) but I would consider doing this in future studies.

**Response to Comment 5:**

The introduction now includes this line about multi-band retrievals

P3, L49:  *However, as is seen in most related studies (e.g. O'Dell et al.*
*(2018)), the major drivers of biases are retrieved surface pressure as well as the retrieved CO2*
*profile shape, and retrieved*
*aerosols and surface albedo contribute much less to the total bias correction (OCO-2 Science*
*Team, 2023).*

I'm not sure you can necessarily conclude that aerosol-surface impacts do not drive changes in multi-band retrievals because surface pressure and CO2 profile shape are the main predictors in the bias correction. An alternative explanation could be that the impact of aerosol scattering may induce changes in the retrieved surface pressure and CO2 profile shape. Both are modified by aerosols - perhaps surface pressure more obviously, but changing the vertical distribution of CO2 can also be related; To first order the CO2 layer jacobians are strongly correlated but differ by scaling factor due to pressure broadening (effectively absorption becomes less efficient at higher altitudes, because the narrow lines are already saturated). In this case unphysical profile shapes could actually be a result of compensating for errors induced by not simulating the correct wavelength-dependent aerosol optical properties between the bands.

It is possibly more informative to look at the change in XCO2 as a function of albedo (e.g. Fig 5. Of Taylor et al. (2023)). In that case at least I think you still do see correlations between the bias correction and surface. I am not saying the multi-band retrievals are useless. My interpretation is that the retrievals are not perfect, aerosol-surface interactions induce some unphysical changes to the retrieved state (which only happens because of the additional light path constraints from the additional bands), and this allows an empirical correction.

Taylor, T. E., O'Dell, C. W., Baker, D., Bruegge, C., Chang, A., Chapsky, L., Chatterjee, A., Cheng, C., Chevallier, F., Crisp, D., Dang, L., Drouin, B., Eldering, A., Feng, L., Fisher, B., Fu, D., Gunson, M., Haemmerle, V., Keller, G. R., Kiel, M., Kuai, L., Kurosu, T., Lambert, A., Laughner, J., Lee, R., Liu, J., Mandrake, L., Marchetti, Y., McGarragh, G., Merrelli, A., Nelson, R. R., Osterman, G., Oyafuso, F., Palmer, P. I., Payne, V. H., Rosenberg, R., Somkuti, P.,

Spiers, G., To, C., Weir, B., Wennberg, P. O., Yu, S., and Zong, J.: Evaluating the consistency between OCO-2 and OCO-3 $XCO_2$ estimates derived from the NASA ACOS version 10 retrieval algorithm, Atmos. Meas. Tech., 16, 3173–3209, https://doi.org/10.5194/amt-16-3173-2023, 2023.

**Response to Comment 5:**

I wasn't suggesting a study of the 1.6 micron CH4 band. The paragraph currently states that with regard to aerosol-surface biases, there is "no reason to assume that the same behavior arises with retrievals from the 1.65 µm window". However the underlying mechanism for the bias is still the same, and because the band is at a shorter wavelength, aerosol and Rayleigh scattering are larger, so if anything it may be slightly worse for the single band retrieval case. I was pointing out the reason why such surface-correlated biases may be lessened is that retrievals from this band often use the CO2 proxy method.

---

## Author Response (AR2)

**Final Response for AMT-2024-145 (technical corrections)**

Peter Somkuti[1,2], Gregory McGarragh[3], Christopher O'Dell[3], Antonio Di Noia[4,5,6], Leif Vogel[4,5,7], Sean Crowell[8], Lesley Ott[2], and Hartmut Bösch[4,5,6]

[1]Earth System Science Interdisciplinary Center, University of Maryland, College Park, MD, USA
[2]Global Modeling and Assimilation Office, National Aeronautics and Space Administration, Goddard Space Flight Center, Greenbelt, MD, USA
[3]Cooperative Institute for Research in the Atmosphere, Colorado State University, Fort Collins, CO, USA
[4]University of Leicester, Leicester, UK
[5]National Centre for Earth Observation, Leicester, UK
[6]Institute of Environmental Physics (IUP), University of Bremen FB1, Bremen, Germany
[7]now at Kaioa Analytics, Mundaka, Viscay, Spain
[8]LumenUs Scientific, Oklahoma City, OK, USA

**Correspondence:** Peter Somkuti (peter.somkuti@nasa.gov)

**1 Regarding comment 2, referee #1, report #2**

**My point was just that for cases where the met model used by CAMS is simulating a cloud where ISCCP is flagging as clear, then the RH in the simulation will be 100%, which will yield an unphysically high AOD due to hygroscopic growth. Provided the model and ISCCP differences are small this won't matter too much, but I would prefer just using**
5 **the cloud data from the ERA5 model to be self consistent in the OSSE. As far as I can tell the main reason for using ISCCP is to simulate realistic observability statistics. My intuition would be to opt for self-consistency above this. It is a relatively minor point (not worth redoing simulations) but I would consider doing this in future studies.**

We now better understand the original comment made and concede that we had not considered the impact on the aerosol particle sizes as influenced by the mismatch between CAMS and ISCCP cloud locations, and thus the mismatch between the humidity
10 profiles; meaning: if CAMS produces a cloud at a location for which ISCCP does not indicate a cloud, the hygroscopic aerosol types would grow to much larger sizes compared to if CAMS did not assume a cloud at that location.

We consider this potential bias to be not significant for the results of our study. The mismatch in humidity profiles would produce higher optical depths (as mentioned in the above comment) to an extent only, as most of the aerosol load is found in the lower levels of the troposphere where the enhancement due to the potential cloud-influenced humidity increase would
15 be minimal or none at all (see the illustrative Figure 4 in the revised manuscript). This possible high-bias in total aerosol optical depth would thus stem from the upper-tropospheric portions. Overall, the bottom-line impact would be that our forward simulations are potentially (slightly) biased towards higher total aerosol optical depths. Of course in terms of the OSSE, the bias is unobservable to the system as we compare against the forward simulations, and not the physical truth.

For the sake of completeness, we chose the ISCCP dataset due to the high spatial resolution, and the fact that we did not have to spend time on creating a spatially consistent cloud scheme that preserves the overall cloud fraction as indicated by the (for example CAMS) model.

**2 Regarding comment 5, referee # 1, report #2**

**[...] I'm not sure you can necessarily conclude that aerosol-surface impacts do not drive changes in multi-band retrievals because surface pressure and CO2 profile shape are the main predictors in the bias correction. An alternative explanation could be that the impact of aerosol scattering may induce changes in the retrieved surface pressure and CO2 profile shape. Both are modified by aerosols - perhaps surface pressure more obviously, but changing the vertical distribution of CO2 can also be related; To first order the CO2 layer jacobians are strongly correlated but differ by scaling factor due to pressure broadening (effectively absorption becomes less efficient at higher altitudes, because the narrow lines are already saturated). In this case unphysical profile shapes could actually be a result of compensating for errors induced by not simulating the correct wavelength-dependent aerosol optical properties between the bands. It is possibly more informative to look at the change in XCO2 as a function of albedo (e.g. Fig 5. of Taylor et al. (2023)). In that case at least I think you still do see correlations between the bias correction and surface. I am not saying the multi-band retrievals are useless. My interpretation is that the retrievals are not perfect, aerosol-surface interactions induce some unphysical changes to the retrieved state (which only happens because of the additional light path constraints from the additional bands), and this allows an empirical correction. [...]**

We were attempting to make a comment on the overall magnitude, suggesting that at the moment, for the case of multi-band greenhouse gas retrievals of this type, the surface pressure- and gas profile shape-related bias correction terms contribute more than the surface-aerosol bias. It was certainly not the intention to suggest that multi-band retrievals are immune to this, so to speak. In order to clarify the point about multi-band retrievals, we changed and extended the paragraph on page 3 to now read the following:

> In most related studies (e.g. O'Dell et al. (2018)), the major drivers of biases are identified as retrieved surface pressure as well as the retrieved $CO_2$ profile shape. The retrieved aerosol optical depth and surface albedo contribute much less to the total bias correction (OCO-2 Science Team, 2023). It seems plausible that surface-aerosol interactions manifest as a different type of bias, for example through interference of surface pressure and aerosol optical depth retrieval. Regardless of the actual mechanism, the utilization of 3-band retrievals from GOSAT, OCO-2 and OCO-3 have made surface-aerosol biases less apparent, and the surface bias is no longer a dominant contribution to the total observer errors.

**3 Regarding comment 6, referee # 1, report #2**

**I wasn't suggesting a study of the 1.6 micron CH4 band. The paragraph currently states that with regard to aerosol-surface biases, there is "no reason to assume that the same behavior arises with retrievals from the 1.65 $\mu$m window". However the underlying mechanism for the bias is still the same, and because the band is at a shorter wavelength, aerosol and Rayleigh scattering are larger, so if anything it may be slightly worse for the single band retrieval case. I was pointing out the reason why such surface-correlated biases may be lessened is that retrievals from this band often use the CO2 proxy method.**

We have modified the paragraph in the Discussion & Conclusions section of the manuscript:

> One can expect that the same behavior arises with retrievals from the 1.65 μm window since the fundamental mechanism that drives the bias is the same, despite the spectroscopic features having different characteristics. The often-used "proxy method" (Frankenberg et al., 2005) provides a solution for some spectral regions, in which a ratio to some reference trace gas is retrieved. In that ration, biases from unaccounted light path modification due to aerosols largely cancel out. This is the retrieval strategy of choice for the MethaneSAT (Chan Miller et al., 2023) mission. In fact, Chan Miller et al. (2023) present first results from the airborne MethaneAir instrument and do not observe any strong surface-related biases.

**References**

50  Chan Miller, C., Roche, S., Wilzewski, J. S., Liu, X., Chance, K., Souri, A. H., Conway, E., Luo, B., Samra, J., Hawthorne, J., Sun, K., Staebell, C., Chulakadabba, A., Sargent, M., Benmergui, J. S., Franklin, J. E., Daube, B. C., Li, Y., Laughner, J. L., Baier, B. C., Gautam, R., Omara, M., and Wofsy, S. C.: Methane retrieval from MethaneAIR using the $CO_2$ Proxy Approach: A demonstration for the upcoming MethaneSAT mission, EGUsphere, 2023, 1–40, https://doi.org/10.5194/egusphere-2023-1962, 2023.

Frankenberg, C., Meirink, J. F., van Weele, M., Platt, U., and Wagner, T.: Assessing Methane Emissions from Global Space-Borne Observa-
55  tions, Science, 308, 1010–1014, https://doi.org/10.1126/science.1106644, 2005.

OCO-2 Science Team: Orbiting Carbon Observatory-2 & 3 - Data Product User's Guide, https://doi.org/10.5067/6719R6CJMH7Y, 2023.

O'Dell, C. W., Eldering, A., Wennberg, P. O., Crisp, D., Gunson, M. R., Fisher, B., Frankenberg, C., Kiel, M., Lindqvist, H., Mandrake, L., Merrelli, A., Natraj, V., Nelson, R. R., Osterman, G. B., Payne, V. H., Taylor, T. E., Wunch, D., Drouin, B. J., Oyafuso, F., Chang, A., McDuffie, J., Smyth, M., Baker, D. F., Basu, S., Chevallier, F., Crowell, S. M. R., Feng, L., Palmer, P. I., Dubey, M., García, O. E., Griffith,
60  D. W. T., Hase, F., Iraci, L. T., Kivi, R., Morino, I., Notholt, J., Ohyama, H., Petri, C., Roehl, C. M., Sha, M. K., Strong, K., Sussmann, R., Te, Y., Uchino, O., and Velazco, V. A.: Improved retrievals of carbon dioxide from Orbiting Carbon Observatory-2 with the version 8 ACOS algorithm, Atmospheric Measurement Techniques, 11, 6539–6576, https://doi.org/10.5194/amt-11-6539-2018, 2018.